# Fine-Tuning of Optical Resonance Wavelength of Surface-Micromachined Optical Ultrasound Transducer Arrays for Single-Wavelength Light Source Readout

**DOI:** 10.3390/mi15091111

**Published:** 2024-08-31

**Authors:** Zhiyu Yan, Cheng Fang, Jun Zou

**Affiliations:** Department of Electrical and Computer Engineering, Texas A&M University, College Station, TX 77843, USA; yan1383@tamu.edu (Z.Y.); fangchengok2007@tamu.edu (C.F.)

**Keywords:** optical ultrasound transducer, 2D transducer array, micromachining, transducer parameter manipulation, photoacoustic computed tomography

## Abstract

This article reports the fine-tuning of the optical resonance wavelength (ORW) of surface-micromachined optical ultrasound transducer (SMOUT) arrays to enable ultrasound data readout with non-tunable interrogation light sources for photoacoustic computed tomography (PACT). Permanent ORW tuning is achieved by material deposition onto or subtraction from the top diaphragm of each element with sub-nanometer resolution. For demonstration, a SMOUT array is first fabricated, and its ORW is tuned for readout with an 808 nm laser diode (LD). Experiments are conducted to characterize the optical and acoustic performances of the elements within the center region of the SMOUT array. Two-dimensional and three-dimensional PACT (photoacoustic computed tomography) is also performed to evaluate the imaging performance of the ORW-tuned SMOUT array. The results show that the ORW tuning does not degrade the optical, acoustic, and overall imaging performances of the SMOUT elements. As a result, the fine-tuning method enables new SMOUT-based PACT systems that are low cost, compact, powerful, and even higher speed, with parallel readout capability.

## 1. Introduction

Photoacoustic computed tomography (PACT), as a hybrid and non-invasive imaging modality, has attracted increasing interest in the past decade [1,2,3]. In PACT, nanosecond laser pulses illuminate biological tissues, causing the local temperature to rise and generating ultrasound through non-radiation relaxation. A single-element transducer or an array (linear or 2D) captures the ultrasound signals, which are decoded and reconstructed as 2D/3D images [4,5,6]. Recently, optical ultrasound transducers (OUTs) have been investigated for ultrasound detection in PACT, owing to their high sensitivity, small footprint, and wide bandwidth. Based on an optomechanical modulation mechanism, OUTs, such as the micro-ring resonator [7] and Bragg grating waveguide [8], can detect Pascal-level acoustic pressure within a bandwidth up to tens of megahertz with the element size even smaller than the ultrasound wavelength. Nevertheless, the variation in the critical dimension of the transducer (e.g., the diameter of a micro-ring resonator), due to the fabrication errors (e.g., non-uniform film thickness or etching rate), makes it challenging to form highly uniform arrays. The low array uniformity requires the operation conditions (e.g., probing optical wavelength) to be adjusted frequently during the signal readout and therefore lowers the data acquisition rate. Recently, we developed a new surface-micromachined optical ultrasound transducer (SMOUT) array based on a Fabry–Perot interferometer, which has high element density, high sensitivity, and ‘wireless’ optical readout capability [9]. More importantly, with new structural designs and fabrication techniques, the optical properties of the entire array (with an active area of several $cm^{2}$) can be made highly uniform, such that all the elements in the entire array can be read out without adjusting the optical wavelength.

As shown in Figure 1a, a SMOUT element is a hollow Fabry–Perot (F-P) cavity formed by two identical DBRs (distributed Bragg reflectors) [9]. Each of the DBRs is made up of a few consecutively deposited silicon dioxide/nitride (SiO/SiN) pairs (as the subfigure ‘multi-layer structure’ in Figure 1a). A layer of low-temperature oxide (LTO) is coated on the top DBR to seal the F-P cavity in a vacuum. When normally incident onto the DBR, the light is reflected at the interfaces between layers with different refractive indices. Multiple reflected beams could cancel each other out through destructive interference at certain wavelengths, which is known as the ORW (optical resonance wavelength) of the F-P cavity. For the reflectance spectrum of the SMOUT element (Figure 1b), the ORW (denoted as $\lambda_{0}$) corresponds to the dip where the reflectance reaches a local minimum. For an F-P cavity constructed by two DBRs (Figure 1a), its ORW is determined by the cavity length (i.e., distance between the top and bottom DBRs) as well as the optical length (production of thickness and refractive index) of thin films that construct the DBRs. When ultrasound waves approach the SMOUT element, the top diaphragm (DBR + LTO) is vibrated, which shifts the cavity length and its reflectance spectrum (Figure 1b). Consequently, $\lambda_{bias}$ (the quadrature point of the spectrum slope) is selected as the interrogation wavelength to maximize the modulation of reflected light intensity by ultrasound as the output signal (Figure 1b).

Although the optical properties of an entire array can be made highly uniform, the ORWs of the SMOUT arrays fabricated on different substrates in one batch or different batches may deviate by a few nanometers or even more due to the unavoidable variations in the fabrication process (e.g., ~nm difference of the deposited film thickness). To compensate for this, a tunable laser is always needed to specifically provide the optimal interrogation wavelength for a SMOUT array. However, the tunable laser is inherently bulky, complex, and expensive, and more importantly, its output power is limited (e.g., ~10 mW), which would be just enough to interrogate a single SMOUT element. As a result, the SMOUT elements over the entire array have to be scanned by the interrogation beam one by one, which significantly limits the overall imaging speed (Figure 2a). One possible solution is to use multiple beams for parallel readout (Figure 2b); however, the required optical power is much higher than the typical output from a tunable laser. In this case, a more powerful light source (e.g., laser diodes (LDs), power ≥ 100 mW) has to be used, but its wavelength is non-tunable. Therefore, the capability to fine-tune the ORWs of different SMOUT arrays toward the output wavelength of the high-power non-tunable light source is essential (Figure 3). Among the non-tunable light sources, the pulsed ones with high repetition rates are more helpful in reducing phase noise and thus improve the signal-to- noise ratio (SNR) and image quality [10].

In this article, we report the fine-tuning of the ORW of 2D SMOUT arrays for enabling a readout with fixed-wavelength light sources. With controllable material deposition onto or subtraction from the top diaphragm of the SMOUT elements, permanent ORW tuning is achieved with sub-nanometer resolution. An optical setup is built with a fixed-wavelength LD for SMOUT characterization and imaging. PACT is also conducted to verify the imaging performance of the ORW-tuned SMOUT array. The experimental results show that the ORW tuning has neglectable effect on the optical, acoustic, and imaging properties of the SMOUT arrays, while making them capable of working with a non-tunable LD as the light source. The ability of fine-tuning and controlling the ORW is a critical step toward the practical applications of SMOUT arrays in PACT, which reduces the imaging system complexity and construction cost by allowing the use of non-tunable high-power light sources. Furthermore, it is also promising to boost the imaging speed with parallel readout capability.

## 2. Methods

### 2.1. Tuning Mechanism

Since the cavity is sealed in a vacuum, the top diaphragm of the SMOUT element is slightly deflected (downward) due to the pressure outside (Figure 4a). Either material coating or etching can change the thickness, overall mechanical stiffness, and, therefore, the deflection of the diaphragm. Coating makes the diaphragm thicker and stiffer (with a higher force constant) and thus decreases its pressure-induced deflection, which increases the gap (L) between the top and bottom DBRs (i.e., the F-P cavity length). For an F-P optical cavity with two reflectors parallel to each other and normal-incident light, the cavity length L has a simplified relationship with the resonance wavelength ORW [11]
(1)2n·L=q·ORW
where n is the refractive index of the medium in the cavity (1 of air) and q is an integer ≥1. As a result, the L increased by coating makes the ORW larger. Oppositely, partially etching the diaphragm makes the diaphragm thinner and softer (with a lower force constant) and thus enlarges its deflection, which decreases the gap between the top and bottom DBRs (i.e., the F-P cavity length) and thus the ORW (Figure 4a). Therefore, the ORW of the SMOUT element can be precisely and permanently increased (from $\lambda_{0}$ to $\lambda_{+}$) and decreased (from $\lambda_{0}$ to $\lambda_{-}$) by coating and etching, respectively (Figure 4b).

### 2.2. Simulation

An FEM (finite element method) model is built only with the Solid Mechanics module of COMSOL Multiphysics 5.3 (Figure 5a) to simulate the ORW tuning by material coating or etching. The cross-section view of the simulated pressure-deflected diaphragm before tuning is shown in Figure 5b. Given the thickness of the removed LTO, the simulated peak deflection of the diaphragm is subtracted from the original cavity length (300 nm) of the SMOUT element. Based on the new cavity length, the reflectance spectrum is calculated with the TMM (transfer-matrix method) algorithm [12]. Figure 5c,d show the simulated spectrum shifts and ORW changes, respectively, indicating a −10 nm ∆ORW and an average rate of −0.8 nm per 100 nm thick LTO removal. The simulation is similar when extra material is coated on the diaphragm. Silicon nitride (SiN) is selected to coat on the diaphragm because of its high Young’s modulus and tuning efficiency. Figure 5e,f show the simulated spectrum shifts and ORW changes, respectively, indicating a +6.2 nm ∆ORW and an average rate of +1.6 nm per 100 nm thick SiN deposition.

### 2.3. Fabrication and Tuning

The fabrication process of the SMOUT array [9] is shown in Figure 6. A 500 µm thick double-sided polished Borosilicate wafer is used as the substrate, on top of which the bottom DBR is deposited (Figure 6a). A layer of sacrificial material is coated onto the bottom DBR by RF (radio frequency) sputtering and then patterned into individual islands by lithography that defines the geometry of each SMOUT element (Figure 6b). Then, the top DBR (identical to the bottom DBR) is deposited by PECVD (plasma-enhanced chemical vapor deposition) to form the cavity structure (Figure 6c). Etching holes are opened through the top DBR by RIE (reactive ion etching) (Figure 6d), and the sacrificial material is removed by wet etching to release the top DBR as a diaphragm (Figure 6e). The SMOUT array is immersed into the (liquid) etching solution. The sacrificial material is gradually etched away, starting from the etching holes on the corner of the top DBR and reaching the center of the diaphragm. Because both the top and bottom DBRs are transparent, the etching process can be clearly monitored under a microscope until all sacrificial material is etched away. After rinsing, the SMOUT array is left in air until the liquid inside the cavity evaporates completely. A layer of LTO (low-temperature oxide) is coated on the top DBR by LPCVD (low-pressure chemical vapor deposition) to seal the cavity in vacuum (Figure 6f). The SMOUT ORW can be tuned by LTO (topmost layer in the diaphragm) wet etching (Figure 6g) or SiN deposition (Figure 6h). ORW tuning by LTO wet etching is demonstrated by a SMOUT array with a 1 × 1 $cm^{2}$ active area, 70 µm element size, 140 µm pitch, and 70 × 70 elements. The array is immersed in a hydrofluoric acid (HF) solution (1:7 dilution ratio to water) for LTO removal, with an ~200 nm/min etching rate under room temperature and agitation (@ 100 rpm). Figure 6i shows the tuned array after LTO wet etching. To prevent the wet etchant leaking into the cavity, the LTO around the etching holes is protected with a photoresist plug (Figure 6j) in advance. The ORW of another identical SMOUT array is simply tuned by SiN PECVD.

The reflectance spectra of five SMOUT elements across the array before/after tuning are characterized with a spectrometer (USB4000, Ocean Optics, Orlando, FL, USA). Figure 7a shows the shifted spectrum of a representative SMOUT element after LTO etching, indicating an overall ∆ORW = −12 nm after 1200 nm thick LTO removal. Figure 7b shows the average and rms (root mean square) of the shifted ORW versus the removed LTO thickness, indicating an average tuning rate of −1.0 nm per 100 nm LTO removal. Figure 7c shows a ∆ORW = +7 nm after 400 nm thick SiN PECVD on the other identical SMOUT array. Figure 7d shows the average and rms of the shifted ORW versus deposited SiN thickness, indicating an average tuning rate of +1.7 nm per 100 nm SiN deposition. Both tuning results are well matched with the simulation results (Figure 5). It can be noticed that the ORW rms/deviation tends to increase with more LTO removal (Figure 7b). This is caused by the non-uniform etching rate in the wet etchant, which tends to accumulate with the increased etching duration. This issue can be addressed by an (dry) etching process (based on plasma or vapor) that has a more uniform etching rate over a large surface area. However, this issue disappears when depositing SiN for ORW tuning (Figure 7d), owing to the high uniformity of PECVD.

## 3. Testing and Results

### 3.1. Testing Setup

To demonstrate the ORW tuning on larger arrays, a 5 × 5-$cm^{2}$ SMOUT array (70 µm element size, 140 µm pitch, and 350 × 350 elements) is first fabricated, and its ORW is tuned from ~815 nm to ~818 nm (by SiN deposition) to optimize its sensitivity at an interrogation wavelength of 808 nm, which is the center wavelength of an LD light source (150-mW, 808-nm, FPL808P, Thorlabs Inc., Newton, NJ, USA). An optical setup is built with the 808 nm LD to characterize the acoustic response performances of the ORW-tuned SMOUT array (Figure 8). The acoustic source is a 0.4 mm thick PZT (lead zirconate titanate) plate driven by an ultrasound pulser/receiver (5072 PR, Olympus, Tokyo, Japan). The PZT plate is placed ~20 mm below the SMOUT array, and water is filled for acoustic coupling. The LD output is first coupled into a single-mode fiber circulator (Precision Micro-Optics, Burlington, MA, USA) and then collimated by a fiber collimator (F810APC-780, Thorlabs Inc., Newton, NJ, USA). The collimated laser beam is focused by a 10× objective lens before illuminating the SMOUT elements. The reflected light from the illuminated SMOUT element is coupled back into the circulator and guided to a photodetector (DET36A, Thorlabs Inc., Newton, NJ, USA). The output voltage signal from the photodetector is filtered and amplified by the pulser/receiver (5072PR, Olympus) and acquired by a DAQ board (ATS9350, AlazarTech, Pointe-Claire, QC, Canada). To better focus and align the interrogation laser spot, light from a halogen light source (HL-2000-LL, Ocean Optics, Orlando, FL, USA) is combined with the interrogation laser by a dichroic mirror (DMLP650, Thorlabs Inc., Newton, NJ, USA). Then, 1% power of the reflected light from the SMOUT array is projected to a CCD camera (MU1603, AmScope, Irvine, CA, USA) by a beam sampler (BSF10-A, Thorlabs Inc., Newton, NJ, USA) to image the laser spot and the surrounding area of the interrogated element. Two motorized scanning stages (NRT100, Thorlabs Inc., Newton, NJ, USA, X and Y stages in Figure 8) are used to scan the interrogation laser spot over the array for data collection.

### 3.2. Uniformity

The 808 nm interrogation beam is scanned over the 12 × 12 $mm^{2}$ central area of the tuned SMOUT array with a step of 0.7 mm (five pitches) to acquire the ultrasound response of each element. A map and histogram of the (peak-to-peak) amplitude of the acquired signals are plotted in Figure 9a,b, respectively. Each block in Figure 9a represents the average signal amplitude from elements within a 0.7 × 0.7 $mm^{2}$ pixel area. More than 90% of the elements have a normalized signal amplitude ranging between 0.7 and 0.9, with a relative standard deviation (STD) of 9.36%. The uniformity of the tuned SMOUT array is comparable with the SMOUT arrays without tuning [9], indicating that the ORW tuning has a neglectable effect on the SMOUT arrays.

### 3.3. Noise Equivalent Pressure and Linearity

Noise equivalent pressure (NEP), defining the minimal detectable acoustic pressure, is an evaluation of the detection sensitivity of a transducer. The NEP of the tuned SMOUT array is characterized by putting a 0.2 mm needle hydrophone (NH0200, Precision Acoustics, Dorchester, UK) at the same location as the SMOUT to measure the acoustic pressure generated from the PZT. The rms pressure level and noise amplitude are measured as 35.2 kPa and 13.7 mV (with 16 averaging), respectively. By acquiring the ultrasound response from nine elements uniformly located in the measured area, the mapping (Figure 10a) indicates an rms NEP of 72.2 Pa within a 10 MHz bandwidth (1 kHz to 10 MHz). In comparison, with a wavelength-tunable laser as the interrogation light source, the NEP is measured as 19.2 Pa, which is close to that of a SMOUT array without tuning (20.7 Pa, reported in [9]). This indicates that the ORW tuning has no significant impact on the SMOUT sensitivity. The NEP is slightly compromised by the LD due to several reasons. First, the monochromaticity of the LD is not as good as that of the tunable laser, causing expansion of the focal spot size from 10 µm to 30 µm due to the dispersion effect. The large beam size lowers the readout optical intensity and the modulation efficiency. In addition, the noise level of the LD is significantly higher than that of the tunable laser, possibly due to insufficient heat management and the increased photodetector thermal noise due to increased optical power. To characterize the linearity of the acoustic response of the tuned array, ultrasound signals from five different SMOUT elements are recorded by reducing the driving voltage of PZT and thus the generated acoustic pressure. The measured ultrasound response is in Figure 10b, showing a high linear responsivity of ~200 mV/kPa.

### 3.4. Frequency Response

Photoacoustic (PA) testing is conducted to evaluate the frequency response of the SMOUT array. A black hair is chosen as the target for its wideband PA generation and high damage threshold. Figure 11a,b show the typical PA signal acquired from a SMOUT element and its frequency spectrum after FFT (fast Fourier transform), respectively. The center frequency and -6dB bandwidth are estimated to be 3.2 MHz and 6.4 MHz (~200%), respectively, which are close to the 3.5-MHz center frequency and 5-MHz 6-dB bandwidth (~140%) before ORW tuning [9]. Again, the wideband detection capabilities of the SMOUT arrays are not changed by the ORW tuning.

### 3.5. Stability

The thermal and temporal stabilities of the tuned SMOUT array are characterized by monitoring the shift of the (normalized) signal amplitude when the array front surface is immersed in water under different ambient temperatures or for a prolonged duration. The ambient temperature is controlled by the water heated by a hot plate and measured by a thermometer. Normalized signal amplitudes of five adjacent elements are recorded at different temperatures (Figure 12a), indicating a relative STD of 6.14% from 25 to 55 °C. Then, the array is continuously immersed in water for 7 days and the normalized signal amplitudes of the five elements are recorded (one per day) as shown in Figure 12b, indicating a relative STD of 4.89%. The stabilities of the tuned SMOUT array are quite close to those of SMOUT without tuning [9], indicating that ORW tuning has a neglectable effect on the stabilities of the SMOUT array.

## 4. Imaging Experiment

### 4.1. Imaging Setup and Data Acquisition

A PACT setup is built to evaluate the imaging capability of the ORW-tuned SMOUT array (Figure 13). The imaging target is placed below the SMOUT array, and water is filled for acoustic coupling. A 532 nm pulsed laser (Nano L90-100, Litron Laser Ltd., Warwickshire, UK, 10 ns pulse duration, 100 Hz repeating rate) is used as the light source for PA excitation. The output laser is collimated and homogenized as a 1 cm diameter spot with 2 mJ/$cm^{2}$. PA waves induced from the target are received by the tuned SMOUT array. The focused Interrogation beam (808 nm, from the LD) is scanned over the SMOUT elements in 1D and 2D arrays, where the laser spot is always aimed at the center of the SMOUT element.

### 4.2. Two-Dimensional Imaging Results

Figure 14a shows a photo of the 2D PACT setup using a strand of black human hair as the imaging target. The hair is fixed along the Y axis and around 9 mm beneath the SMOUT array. The SMOUT array is placed within the X-Y plane, and the front surface is facing toward the hair. The 808 nm laser spot is 1D scanned along the X axis with a 0.14 mm increment (every element) and a 35 mm range (marked as the yellow dashed arrow in Figure 14a) to interrogate the PA signals. After bandpass filtering (1–10 MHz) and Hilbert transformation, a 2D B-scan image is reconstructed as shown in Figure 14b, where the ‘arc’ depicts the spatial distribution of the hair-induced PA wavefront within the X-Z plane. Based on Figure 14b and the k-space method [13], the cross-section image of the hair in the X-Z plane is further reconstructed (Figure 14c). Since the thin hair (~100 µm in diameter) can be treated as an impulse input in the X-Z plane, the intensity profile (Figure 14c) can be regarded as the PSF (point spread function) and used to characterize the spatial resolution of the PACT setup. Based on the FWHM (full width at half maximum) of the Z-axis profile (Figure 14d), the axial resolution is estimated to be 330 µm, which is mainly determined by the center frequency and bandwidth of the SMOUT element [14]. Similarly, based on the X-axis profile, the lateral resolution is estimated to be 420 µm at a depth of 9 mm (Figure 14e). Both the axial and lateral resolutions are comparable with those of the SMOUT array without ORW tuning [9] as well as the piezoelectric transducer arrays operating at similar acoustic frequencies [15]. Note that the lateral resolution can be degraded at a larger depth due to the narrower viewing angle.

### 4.3. Three-Dimensional Imaging Results

A 3D PACT setup is built to further investigate the 3D imaging performance of the ORW-tuned array upon targets within certain FoVs (fields of view) and depths. As shown in Figure 15a, three pencil-lead dots fixed on three parallel 532 nm transparent optical fibers are used as the targets. Each dot is approximately 1 mm long and 0.5 mm in diameter. The spacing between two adjacent dots is 10 mm (along X and Y) and 5 mm (along Z). The top pencil-lead dot is ~10mm beneath the SMOUT array, which is fixed within the X-Y plane. By scanning the interrogation light spot over the central 30 mm × 30 mm (X-Y plane) region of the tuned SMOUT array, a 3D image is reconstructed (Figure 15b) under the same protocol as that of Figure 14c, which clearly shows the location/shape of the targets. The peak contrast-to-noise ratio (CNR) is measured as 60.1 dB, which is comparable to that obtained with the SMOUT array without ORW tuning [9].

## 5. Conclusions and Discussion

In conclusion, we have demonstrated the method of fine-tuning the ORW of the SMOUT array by etching or depositing the top diaphragm and thus changing its thickness and mechanical stiffness. As a result, the ORW of the SMOUT arrays fabricated on different substrates in one batch or different batches can be precisely controlled to enable interrogation with the non-tunable light sources. The experimental results show that such tuning can be performed without degrading the optical, acoustic, and overall imaging performances of the SMOUT arrays. Therefore, the fine-tuning method could be useful in developing new SMOUT-based PACT systems with fixed-wavelength high-power light sources even for the parallel signal readout of multiple SMOUT elements. In vivo small-animal functional imaging, such as mouse brain imaging at a video rate that shows the real-time hemodynamics, will then become possible. In the future, interrogation light sources with higher stability and photodetectors with better noise and fluctuation rejection will be investigated to reduce system noise and further improve imaging quality.

## Figures and Tables

**Figure 1 micromachines-15-01111-f001:**
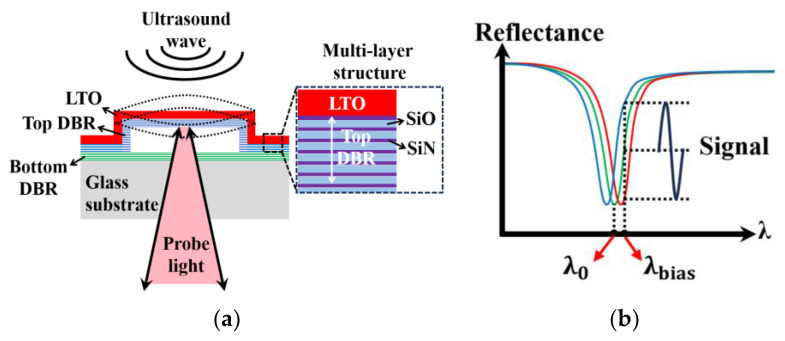
(**a**) The cross-section of a SMOUT element, the top DBR diaphragm (constructed by multi-layer SiN/SiO) of which is vibrated by the impinging ultrasound wave; (**b**) the SMOUT reflectance spectrum shifted by the top diaphragm vibration, where $\lambda_{0}$ and $\lambda_{bias}$ are the ORW and interrogation wavelength, respectively.

**Figure 2 micromachines-15-01111-f002:**
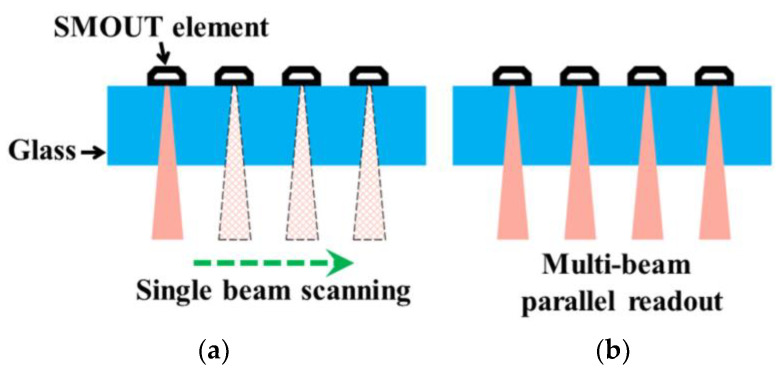
SMOUT readout (**a**) in serial with a single beam (with low optical power) and (**b**) in parallel with multiple beams (with high overall optical power).

**Figure 3 micromachines-15-01111-f003:**
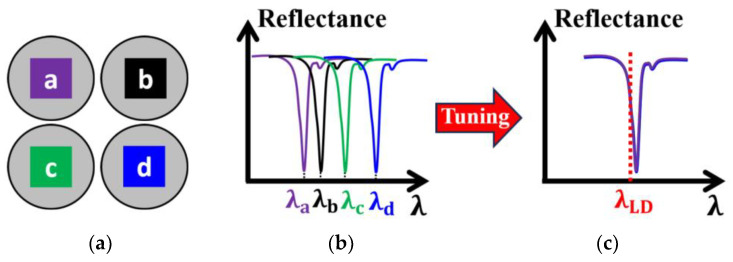
(**a**) Four representative SMOUT arrays (a, b, c, and d) fabricated on different substrates in one batch or different batches; (**b**) reflectance spectra and ORWs ($\lambda_{a}$, $\lambda_{b}$, $\lambda_{c}$, and $\lambda_{d}$ for a, b, c, and d array, respectively) before the tuning; (**c**) uniform reflectance spectra and ORWs after the tuning ($\lambda_{LD}$: the output wavelength of the high-power non-tunable LD).

**Figure 4 micromachines-15-01111-f004:**
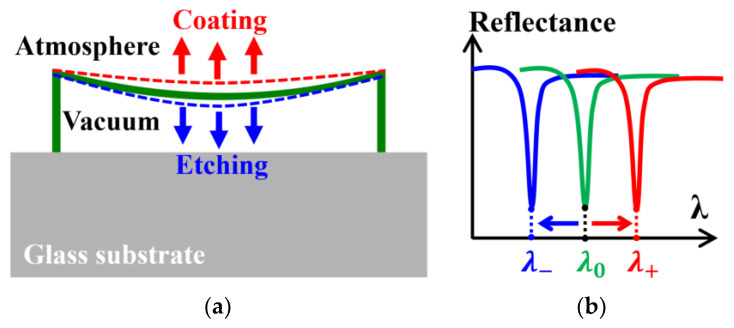
(**a**) Pressure-induced deflection and cavity length changes and (**b**) the corresponding ORW shifts due to coating or etching the top diaphragm of the SMOUT element.

**Figure 5 micromachines-15-01111-f005:**
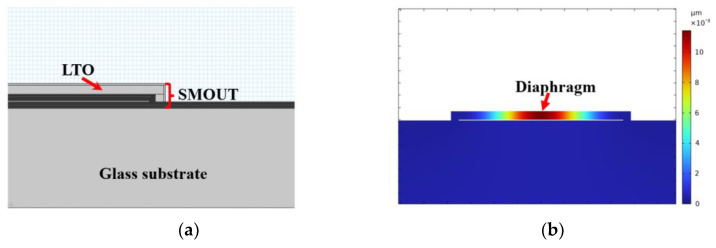
COMSOL simulation of ORW tuning by coating or etching the top DBR diaphragm: (**a**) the FEM model built in COMSOL Multiphysics; (**b**) cross-section view of the simulated pressure-deflected diaphragm before tuning; (**c**) spectrum shifts with a −10 nm overall ∆ORW and (**d**) an average tuning rate of −0.8 nm per 100 nm by LTO removal; (**e**) spectrum shifts with a +6.2 nm overall ∆ORW and (**f**) an average tuning rate of +1.6 nm per 100 nm by SiN deposition.

**Figure 6 micromachines-15-01111-f006:**
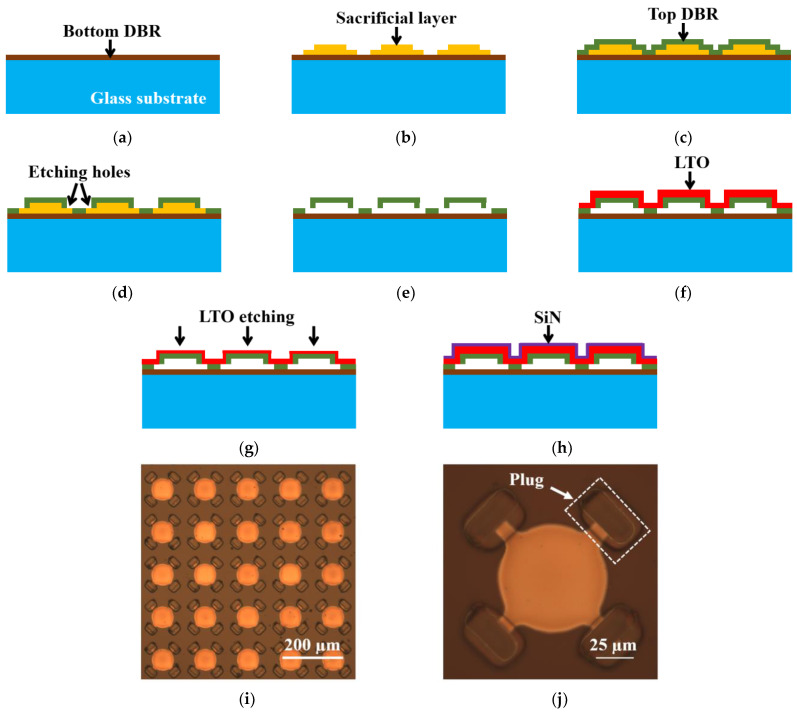
Fabrication process of a SMOUT array: (**a**) PECVD of the bottom DBR on a glass substrate; (**b**) RF sputtering and lithography patterning of a sacrificial layer on the bottom DBR; (**c**) PECVD of the top DBR; (**d**) etching holes opened through the top DBR by RIE to partially expose the sacrificial layer; (**e**) wet etching of the sacrificial layer to release the top DBR as the diaphragm, which is still linked to the bottom DBR at the edges; (**f**) LPCVD of LTO for cavity sealing in a vacuum; ORW tuning by (**g**) LTO etching or (**h**) SiN PECVD; (**i**) a photo (under the microscope) of the tuned SMOUT array after LTO wet etching; and (**j**) a zoom-in view of one SMOUT element with four etching holes at the corners.

**Figure 7 micromachines-15-01111-f007:**
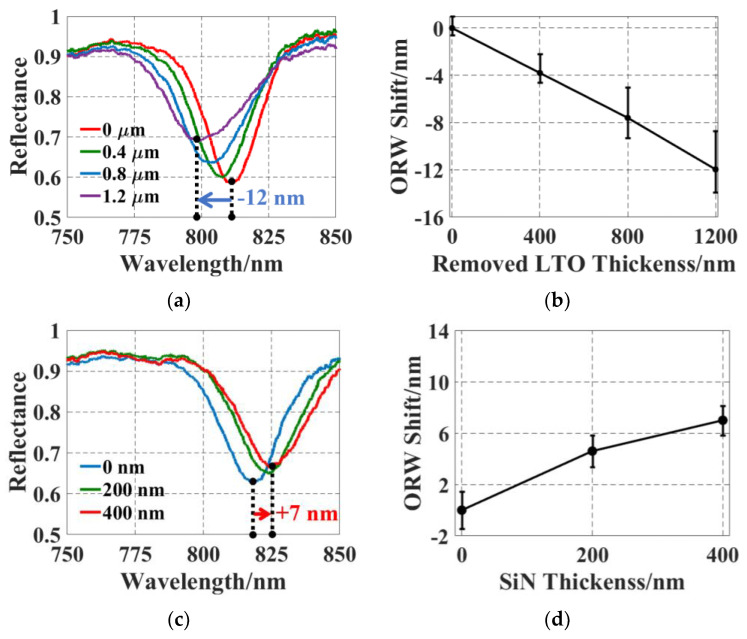
(**a**) Reflectance spectrum shifts with overall −12 nm ∆ORW and (**b**) an average tuning rate of −1.0 nm per 100 nm by LTO removal. (**c**) The spectrum shifts with overall +7 nm ∆ORW and (**d**) an average tuning rate of +1.7 nm per 100 nm by SiN deposition. The error bars in (**b**,**d**) indicate the ORW deviation among the five tested elements.

**Figure 8 micromachines-15-01111-f008:**
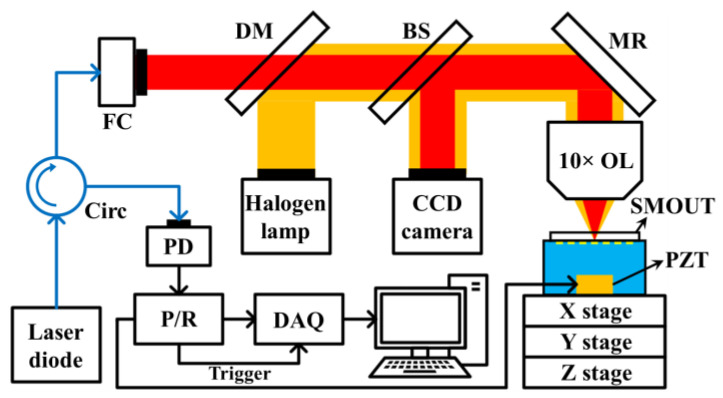
The LD-based setup for characterizing the ORW-tuned SMOUT array (Circ: fiber circulator; FC: fiber collimator; DM: dichroic mirror; BS: beam sampler; PD: photodetector; P/R: pulser/receiver; DAQ: data acquisition board; OL: objective lens; MR: mirror).

**Figure 9 micromachines-15-01111-f009:**
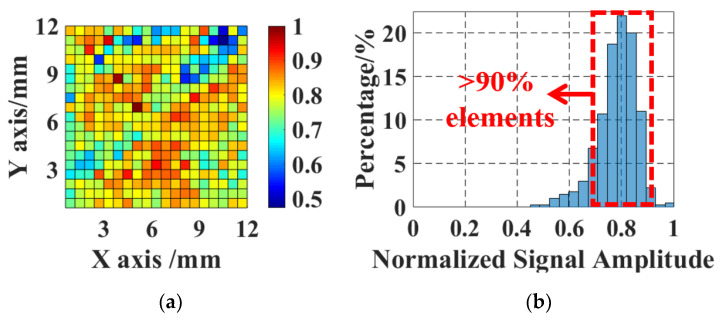
(**a**) The map and (**b**) corresponding histogram showing spatial distribution of signal amplitude from elements in the 1.2 × 1.2 $cm^{2}$ center region of the SMOUT array.

**Figure 10 micromachines-15-01111-f010:**
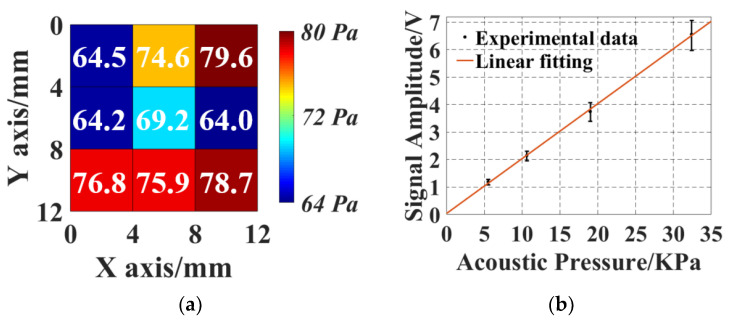
(**a**) NEP map of nine SMOUT elements within the central 12 × 12 $mm^{2}$ area of the ORW-tuned array; (**b**) the measured linearity of the ultrasound response of the SMOUT array. Error bars in (**b**) indicate the deviation in the signal amplitude among different elements.

**Figure 11 micromachines-15-01111-f011:**
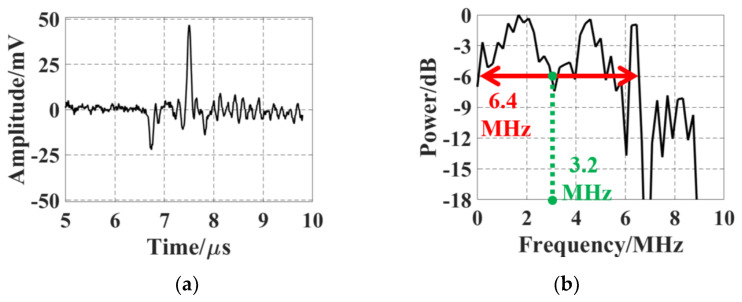
(**a**) A typical PA response of an element from black human hair as the target; (**b**) the FFT spectrum of the PA signal in (**a**) to characterize the frequency response of the tuned SMOUT array.

**Figure 12 micromachines-15-01111-f012:**
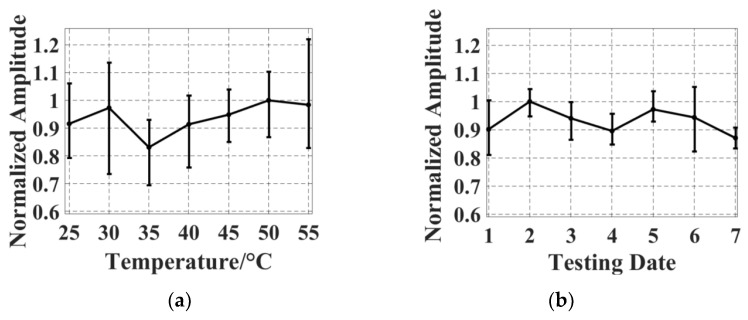
Fluctuations in the normalized signal amplitude received by the tuned SMOUT array due to (**a**) ambient temperature drift from 25 °C to 55 °C and (**b**) continuous immersion in water for one week. Error bars indicate the deviation in the signal amplitude among different elements.

**Figure 13 micromachines-15-01111-f013:**
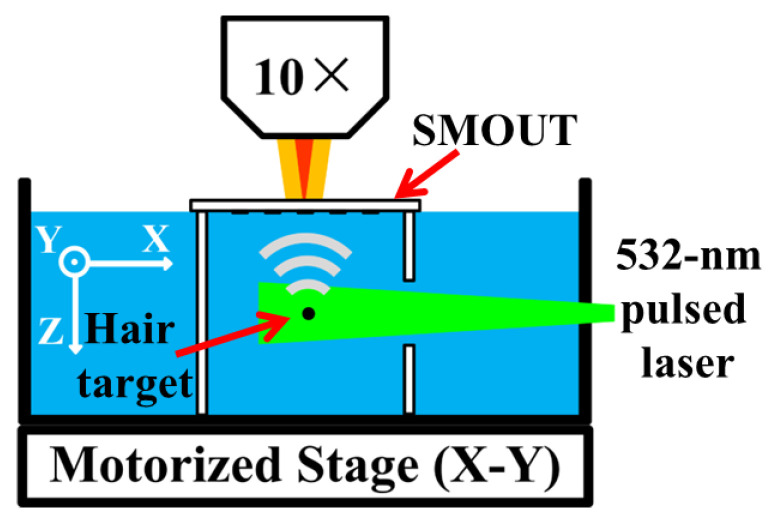
Imaging setup for PACT experiments.

**Figure 14 micromachines-15-01111-f014:**
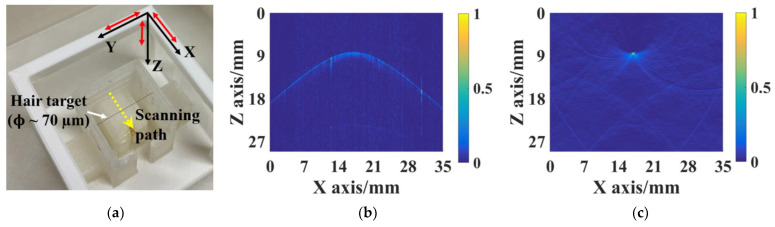
(**a**) Imaging setup for 2D PACT with a strand of black hair as the target, where the scanning path of the interrogation laser spot is marked by the yellow dashed arrow; (**b**) reconstructed B-scan image from the 1D scanning; (**c**) reconstructed 2D image of the black hair, the profile of which along (**d**) the Z and (**e**) X axes is used to evaluate the axial and lateral resolution, respectively. Red arrows in (**a**) represent 10 mm length in the corresponding axes.

**Figure 15 micromachines-15-01111-f015:**
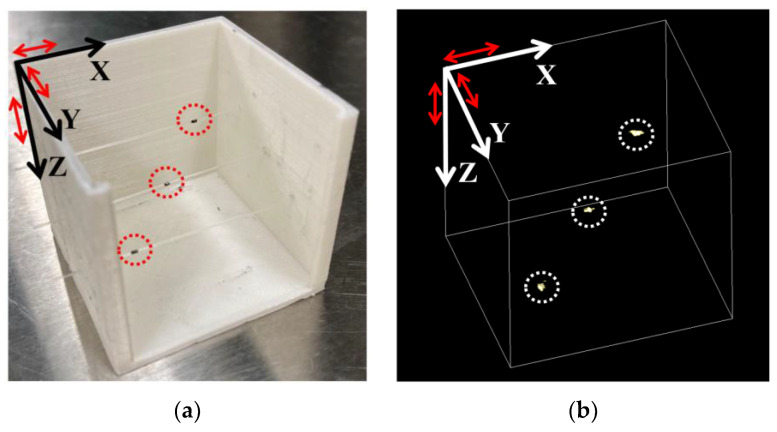
(**a**) Three dot-shaped pencil leads as imaging targets (marked by red dashed circles) for 3D PACT; (**b**) the reconstructed 3D image of the targets (marked by white dashed circles) after 2D scanning the central 3 × 3 $cm^{2}$ region of the tuned SMOUT array. Red arrows in (**a**,**b**) indicate 10 mm length in the corresponding axes.

## Data Availability

Data underlying the results presented in this paper are not publicly available at this time but may be obtained from the authors upon reasonable request.

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
