# Peer review of "Fine-Tuning of Optical Resonance Wavelength of Surface-Micromachined Optical Ultrasound Transducer Arrays for Single-Wavelength Light Source Readout"

_micromachines, 2024, doi:10.3390/mi15091111_

Round 1

Reviewer 1 Report

Comments and Suggestions for Authors

This manuscript is suitable for publication in Micromachines. However, a minor revision of this manuscript should be required before publication.

1. The dry etching process used needs further clarification. Is it ICP process?

2. Further explanation is needed for the COMSOL simulation process, including which physical fields were used and what kind of coupling was performed?

3. Please further describe the potential application value of this research.

Author Response

Reviewer 1

This manuscript is suitable for publication in Micromachines. However, a minor revision of this manuscript should be required before publication.

  1. The dry etching process used needs further clarification. Is it ICP process?

Response: Thank the reviewer for pointing this out. The ‘dry etching process’ can be based on plasma or vapor. It is not necessary to be an ICP process, though ICP is helpful for boosting up the density of the reactive ions and thus increasing the etching rate.

Action: The corresponding sentence has been updated as below. Please see the highlighted text in Line 207-208.

‘This issue can be addressed by a (dry) etching process (based on plasma or vapor) which has a more uniform etching rate over a large surface area.’

  1. Further explanation is needed for the COMSOL simulation process, including which physical fields were used and what kind of coupling was performed?

Response: Thank the reviewer for the suggestion. Only ‘Structural Mechanics à Solid Mechanics (solid)’ module is used for the COMSOL simulation, so no coupling among multiple physical fields is performed. As Figure 5a, only the LTO thickness is changed to mimic the material deposition onto or subtraction from the top diaphragm, whose displacement is provided by the simulation result.

Action: The 1st sentence in Section 2.2 has been expanded (as below) where the updates are highlighted. Please see the highlighted text in Line 134.

‘An FEM (finite element method) model is built only with Solid Mechanics module of COMSOL Multiphysics (Figure 5a) to simulate the ORW tuning by material coating or etching.’

Figure 5a has been updated by marking the layer of LTO on top of the diaphragm. Please see the highlighted figure in Line 148.

  1. Please further describe the potential application value of this research.

Response: Thank the reviewer for the suggestion.

Action: The last sentence of Introduction has been updated (as below). Please see the highlighted text from Line 106 - 110.

‘The ability of fine-tuning and controlling the ORW is a critical step toward the practical applications of SMOUT arrays in PACT, which reduces the imaging system complexity and construction cost by allowing the use of non-tunable high-power light sources. Furthermore, it is also promising to boost up the imaging speed with parallel readout capability.’

Reviewer 2 Report

Comments and Suggestions for Authors

In this article, the authors presented a method for fine-tuning the optical resonance wavelength of SMOUT arrays, enabling their use with non-tunable light sources in photoacoustic imaging. By precisely adjusting the top diaphragm, the ORW is controlled without compromising performance. The manuscript is well-written, and the scientific outcomes are easy to follow. However, the author might consider addressing the following comments to further enhance it.

1.      It would be beneficial for readers if the author could add a few sentences explaining how the coating and etching on the diaphragm influence the ORW of the SMOUT, both in terms of positive and negative shifts.

2.      Could the author elaborate on the removal of sacrificial materials through wet etching? Additionally, it would be helpful to understand how the complete removal of these materials is ensured.

3.      If the sacrificial materials are successfully removed, could the author clarify how the top DBR layers remain suspended in the air, as shown in Figure 6E?

4.      The fabrication section might benefit from additional details, including a more thorough explanation of the fabrication protocols and parameters used.

Author Response

Reviewer 2

In this article, the authors presented a method for fine-tuning the optical resonance wavelength of SMOUT arrays, enabling their use with non-tunable light sources in photoacoustic imaging. By precisely adjusting the top diaphragm, the ORW is controlled without compromising performance. The manuscript is well-written, and the scientific outcomes are easy to follow. However, the author might consider addressing the following comments to further enhance it.

  1. It would be beneficial for readers if the author could add a few sentences explaining how the coating and etching on the diaphragm influence the ORW of the SMOUT, both in terms of positive and negative shifts.

Response: Thank the reviewer for pointing this out.

Action: We have expanded the explanation of the tuning mechanism in Section 2.1 (as below). Please see the highlighted text in Line 117-124.

‘Since the cavity is sealed in vacuum, the top diaphragm of the SMOUT element is slightly deflected (downward) due to the pressure outside (Figure 4a). Either material coating or etching can change the thickness, overall mechanical stiffness, and therefore the deflection of the diaphragm. Coating makes the diaphragm thicker and stiffer (with higher force constant) and thus decreases its pressure-induced deflection, which increases the gap (L) between the top and bottom DBRs (i.e., the F-P cavity length). For an F-P optical cavity with two reflectors parallel to each other and normal-incident light, the cavity length L has a simplified relationship with the resonance wavelength ORW [11]

                                                          2n·L = q·ORW                                                           (1)

where n is the refractive index of the medium in the cavity (1 of air) and q is an integer 1. As a result, the L increased by coating makes the ORW larger. In opposite, partially etching the diaphragm makes the diaphragm thinner and softer (with lower force constant) and thus enlarges its deflection, which decreases the gap between the top and bottom DBRs (i.e., the F-P cavity length) and thus the ORW (Figure 4a). Therefore, the ORW of the SMOUT element can be precisely and permanently increased (from λ0 to λ+) and decreased (from λ0 to λ-) by coating and etching, respectively (Figure 4b).’

  1. Could the author elaborate on the removal of sacrificial materials through wet etching? Additionally, it would be helpful to understand how the complete removal of these materials is ensured.

Response: Thank the reviewer for the great suggestion. The SMOUT array is immersed into the (liquid) etching solution. The sacrificial material is gradually etched away, starting from the etching holes on the corner of the top DBR and reaching the center of the diaphragm. Because both top and bottom DBRs are transparent, the etching process can be clearly monitored under a microscope until all sacrificial material is etched away. After rinsing, the SMOUT array is left in air until the liquid inside the cavity evaporates completely.

Action: We have added the above explanation into the manuscript. Please see the highlighted text in Line 166-172.

  1. If the sacrificial materials are successfully removed, could the author clarify how the top DBR layers remain suspended in the air, as shown in Figure 6E?

Response: Thank the reviewer for pointing this out. Figure 6e only shows the cross-section of the etching holes between the top DBR layers and the substrate. After the sacrificial materials are removed, the top DBR layers are still linked to the bottom DBR at the edges (see Figures 6i & 6j, where four etching holes are at the corners of each SMOUT element).

Action: We have added the above explanation into the captions of Figure 6e and 6j (as below). Please see the highlighted text in Line 191-192, & 194.

‘(e) wet etching of sacrificial layer to release the top DBR as the diaphragm, which is still linked to the bottom DBR at the edges;’

‘(j) the zoom-in view of one SMOUT element with four etching holes at corners’

  1. The fabrication section might benefit from additional details, including a more thorough explanation of the fabrication protocols and parameters used.

Response: Thank the reviewer for the suggestion. Details about the fabrication process has been reported in our previous publication (see Ref. [9]). Here, we tend to neglect them to avoid redundant information.

Action: We have added Ref. [9] in the first sentence of Section 2.3. Please see the highlighted text in Line 158.
